# Heart and brain: Change in cardiac entropy is related to lateralised visual inspection in horses

Martina Felici [1,2], Adam R. Reddon [3]*, Veronica Maglieri[4], Antonio Lanatà [5], Paolo Baragli[1,6]

**1** Department of Veterinary Sciences, University of Pisa, Pisa, Italy, **2** Department of Agricultural and Food Sciences, University of Bologna, Bologna, Italy, **3** School of Biological and Environmental Sciences, Liverpool John Moores University, Liverpool, United Kingdom, **4** Department of Biology, Unit of Ethology, University of Pisa, Pisa, Italy, **5** Department of Information Engineering, University of Florence, Florence, Italy, **6** Bioengineering and Robotic Research Centre "E. Piaggio", University of Pisa, Pisa, Italy

\* A.R.Reddon@ljmu.ac.uk

**Data Availability Statement:** Data are available as a supplementary file.

**Funding:** The author(s) received no specific funding for this work.

## Abstract

Cerebral lateralisation is the tendency for an individual to preferentially use one side of their brain and is apparent in the biased use of paired sensory organs. Horses vary in eye use when viewing a novel stimulus which may be due to different physiological reactions. To understand the interplay between physiology and lateralisation, we presented a novel object (an inflated balloon) to 20 horses while electrocardiogram traces were collected. We measured the amount of time each horse looked at the balloon with each eye. We calculated 'sample entropy' as a measure of non-linear heart rate variability both prior to and during the stimulus presentation. A smaller drop in sample entropy values between the habituation phase and the sample presentation indicates the maintenance of a more complex signal associated with a relaxed physiological state. Horses that spent longer viewing the balloon with their left eye had a greater reduction in sample entropy, while time spend looking with the right eye was unrelated to the change in sample entropy. Therefore, the horses that exhibited a greater reduction in sample entropy tended to use their right hemisphere more, which may take precedence in emotional reactions. These results may help to explain the variation in lateralisation observed among horses.

## Introduction

The two cerebral hemispheres have functional differences, particularly in the sensory systems [1]. This feature has behavioural and physiological consequences [2], and possible practical implications in domestic animals [3, 4]. Within populations, animals may differ in both the strength and direction of lateralisation, which may correlate with cognition and behaviour [5]. Interindividual variation in lateralisation may be explained by frequency-dependent selection acting on the resultant behavioural or cognitive biases [6].

The activity of the two hemispheres may differ depending on the situation in which the animal finds itself. The left hemisphere appears to be involved in the analytical categorization of

**Competing interests:** The authors have declared that no competing interests exist.

stimuli, whereas the right hemisphere is predominant in the response to threat, predator detection, escape, and in the processing of negatively valent stimuli [4]. Therefore, the right hemisphere seems to dominate when a strong reaction can be triggered by a stimulus that has attracted the animal's attention [4] for example, a stress response. Eye use, and therefore hemispheric bias may be useful for non-invasively detecting stress in domestic animals [3]. Based on these findings it seems that lateralised behaviours may be related to the activity of the cerebral hemispheres when animals experience an emotional reaction to environmental stimuli. However, to fully understand the animal's emotionality, a measure of physiological involvement is required.

Emotions can be defined as inner states of the organism that build on previous experiences to operate as a predictive framework for reacting to the environment [7, 8]. However, this definition does not explain how emotions are formed within the organism. Regardless of how emotions originate, it is accepted that there is an interaction between the central nervous system and peripheral activity of the autonomic nervous system (ANS) in determining the emotional response [7].

In humans, studies on central nervous system activity during emotional reactions have mostly been conducted using functional neuroimaging and electroencephalogram (EEG). These investigations have allowed the heart-brain axis to be analysed mathematically in humans [9]. Despite some promising results in the application of 4-lead EEG in horses [10, 11], these measures can be challenging to collect in domestic animals, but behaviour can be assessed as a manifestation of central nervous system involvement for example as a function of brain lateralisation. On the other hand, once EEG traces are acquired, it is relatively straightforward to analyse ANS activity in animals using heart rate variability (HRV).

Heart rate variability represents variation in the temporal distance (in milliseconds) between consecutive electrocardiogram R-peaks which signify the rhythmic oscillations in the cardiac function regulatory system, influenced by the ANS. Hence, by analysing HRV it is possible to study the influence of the ANS on cardiac activity [12, 13]. Several methods have been developed to analyse HRV. The time and frequency domain methods fully characterise a biological signal (such as an ECG) only if it has Gaussian distribution [12]. However, the mammalian nervous system functioning is complex and non-linear [13]. That is why non-linear HRV parameters which characterise the cardiovascular system as a time-varying dynamic system, have been developed [14, 15]. Among them, cardiac entropy is a non-linear parameter that quantifies the disorder of fluctuations in a time series of a physiological signal, such as an ECG recording [13, 16]. High entropy values indicate a greater irregularity in the signal [16], which is usually associated with normal physiological processes. Conversely, lower entropy values are associated with homeostasis imbalance [17]. A nonlinear analysis of physiological signals is required to characterize functional brain-heart interplay [9].

To elicit emotions in animals, stress tests have been widely used, since the expression of negative emotions is usually easier to quantify than positive emotions [18]. The sudden or unexpected presentation of an unfamiliar stimulus may induce a stress response. In animals with laterally placed eyes, for example, horses or other ungulates, the Novel Object Test (NOT) in which an unfamiliar object appears suddenly within a familiar environment, may be used to analyse the animals' emotionality by measuring their preference to view the object with the right or left eye. Horses that perceive the novel object as more threatening may inspect the stimulus with a greater preference for left monocular vision and thus the right cerebral hemisphere, whereas horse that finds the object more interesting or curious may prefer their right eye and left hemisphere to view it [1, 19, 20]. Recently, lateralised behaviours have been linked with heart rate in horses [21] and dogs [22]. As result, variation in the emotional reaction elicited by the object may underpin variation in the laterality of eye use. Alternatively, stimuli

which happen to be detected in one eye and hence are processed first in one hemisphere over the other may lead to different emotional responses to the same stimulus. In this case, the eye use drives the emotional response rather than the other way around.

In a previous study using the NOT in horses, we found that out of 77 horses, 30 individuals preferred to use their right eye to view the object, 26 used their left, and 21 showed a weak or absent preference [1]. In the current study, we aim to correlate eye use during the NOT with changes in cardiac sample entropy to test the hypothesis that variation in emotional response to the stimulus is related to variation in eye use. We predict that horses which use their left eye/right hemisphere will show a greater drop in sample entropy, indicating a more stressed or activated emotional state, than horses which primarily use their right eye/left hemisphere.

## Materials and methods

Twenty horses (11 females; 9 geldings) from two stables (10 horses from each stable) aged 6.05 ± 2.80 (mean ± S.D.) were selected for this study. One of the two stables housed horses for gallop racing, with horses being trained daily and managed exclusively indoors, except for training time, and the other was recreational, with horses being retired or trained three times a week and managed both indoors and outdoors (paddocks). The husbandry protocols were similar in both stables, with the administration of hay and concentrate three times per day and ad libitum water. All subjects included in this study were in good health. The clinical ophthalmological examination showed no deficits in visual function. All horse owners gave written consent for each horse to participate in the study.

All subjects underwent a standardized novel object test (NOT), as in our previous study [1]. Briefly, the NOT consisted of the sudden and unexpected appearance of an orange balloon, 25 cm Ø when inflated. The NOT presentation apparatus consisted of a custom-built metal panel (40 x 40 cm) with a hole in the centre (approximately 3 cm Ø, from which the balloon appeared), supported at the sides by two metal panels keeping it at an inclination of 30˚. A video camera (GoPro 7 Hero White, GoPro, California, USA) was placed at the level of a second hole, 15 cm above the balloon position. For further details see [1]. After 5 minutes of habituation, the balloon was inflated remotely, using compressed air (RevolutionAIR Tank, FNA spa, Robassomero, Italy), by an experimenter outside the stall, through a 10 m tube, at 2 bar pressure for 6 seconds. The balloon remained inflated for 5 minutes while the horse's behaviour was recorded.

To evaluate horses' cardiac reactions, 5 minutes of electrocardiograms (ECG) were recorded at a sampling rate of 250 Hz during both the baseline habituation phase and the NOT. ECG traces were collected non-invasively, through an elastic belt integrated with two textile electrodes (Smartex Srl, Navacchio, Italy) connected to a recording system (Biopac, Goleta, USA). The two electrodes (about 13x5 cm) were placed in a modified base-apex derivation, without shaving the coat and by adding electroconductive gel (Farmacare, San Pietro in Casale, Italy). ECG signals were transmitted from the electronic control unit in Bluetooth Low Energy (BLE) mode to a mobile device. It was therefore possible to remotely monitor ECG traces in real time. For further details on the equipment see [23, 24].

Video analysis was conducted frame by frame, with the VLC media player program (VideoLAN). The amount of time each horse spent looking at the balloon in monocular fixation was coded throughout the 5-minute trial. In this study, the horse was considered engaged in monocular investigation when only one eye was visible from the perspective of the balloon (i.e., when only one eye was visible on the video; Fig 1). To allow for direct comparison of eye use data with a different set of horses used in a previous study [1], we calculated a lateralisation

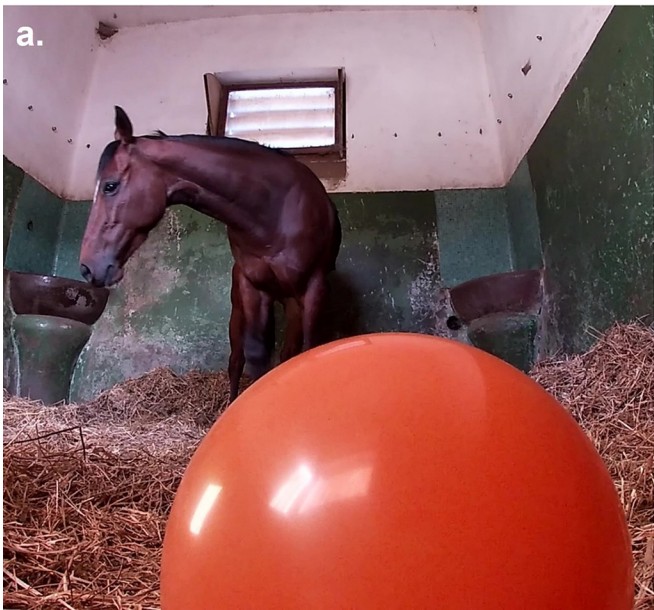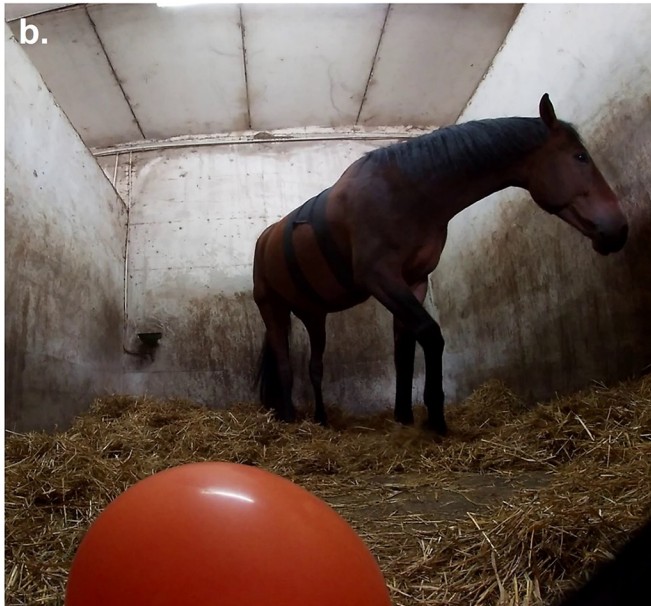

**Fig 1.** Monocular visual inspection on the left (A) and right (B) sides during the novel object test.

index (LI = [right-left] / [right + left]) for each horse's time spent investigating the balloon with right vs. left eye.

ECG traces were imported on a computer using Launch Smart Scope software (Smartex Srl, Navacchio, Italy). Then ECG's R-peaks were detected and corrected through Kubios software (Kubios Oy, Kuopio, Finland). A specific algorithm has been implemented for movement arte-fact removal by ad-hoc analysis (Bioengineering group, Department of Information Engineering, University of Florence) in MATLAB Environment 2021 (MathWorks, Natick, USA). ECG traces were processed, and cardiac Sample Entropy (SampEn) was computed. We calculated ΔSampEn based on the difference between the SampEn value from the baseline measurement and the value collected during the NOT using the following formula: ΔSampEn = (SampEn Test/SampEn Baseline)*100. A ΔSampEn value of 100 indicates no change between the baseline measurement and the test phase, values <100 indicate a decrease in SampEn during the NOT, and values >100 indicate an increase in the SampEn during the NOT relative to baseline.

We calculated the Pearson correlation coefficient between ΔSampEn and each of the left and right eye inspection time. We compared the sexes and stables of origin for both eye use and ΔSampEn using Welch's t-tests. Because we did not restrain the horses during the trials, the horses may have begun in any position relative to the novel object, which could affect their tendency to use one eye or the other. Although we cannot rule out this possibility with our design, we compared left, right, and frontal starting horses in their use of the left or right eye during the task as well as in their ΔSampEn using one-way ANOVAs to explore the degree to which starting position affected the variables of interest. In all analyses, the time spent looking with the left or right eye was square root transformed to account for right skew of the data. ΔSampEn was normally distributed and both left and right eye time were normally distributed following the square root transformation. Analysis and visualisation were conducted using R 4.2.3 (R-Core-Team, 2023), RStudio v.2023.03.0., and SPSS (IBM) v.27 for Macintosh.

## Results

Horses that spent more time viewing the balloon with their left eye had a greater drop in their SampEn (greater reduction in the disorder of fluctuations of the R-R waves time series; ΔSampEn) during the NOT compared to the baseline measure (r = -0.57, p = 0.009; Fig 2A). Conversely, there was no relationship between ΔSampEn and the time spent viewing the balloon with their right eye (r = -0.079, p = 0.74; Fig 2B). There was no difference between the sexes or the stables in ΔSampEn (sex: $t_{17.59}$ = 1.15, p = 0.27; stable: $t_{12.94}$ = 1.11, p = 0.29), left eye (sex: $t_{11.82}$ = 0.22, p = 0.83; stable: $t_{15.61}$ = 0.10, p = 0.93) or right eye (sex: $t_{10.18}$ = 0.90, p = 0.39; stable: $t_{16.80}$ = 1.20, p = 0.25) inspection time. Starting position of the horse (left eye facing the balloon, right eye facing the balloon, both eyes facing the balloon at the onset of the trial) did not affect ΔSampEn ($F_{2,17}$ = 0.44, p = 0.65), left eye ($F_{2,17}$ = 0.42, p = 0.66), or right eye ($F_{2,17}$ = 0.48, p = 0.63) inspection time.

Examination of the laterality index showed that out of 20 horses, 7 preferred (laterality index ≥ 0.5 = ≥75% of the time spent in monocular fixation) to use their right eye, 5 preferred their left eye, and 8 showed weak or absent preference between eyes when viewing the balloon. This distribution of eye use preferences qualitatively matches a larger sample of horses tested in the same behavioural paradigm in a previous study [1].

## Discussion

The horses who spent more time looking at the balloon with their left eye showed a greater reduction in sample entropy during the novel object task. The time spent inspecting the balloon with the right eye was unrelated to the change in SampEn. Most horses showed a decrease in SampEn from the baseline recording to the NOT, suggesting that the balloon presentation

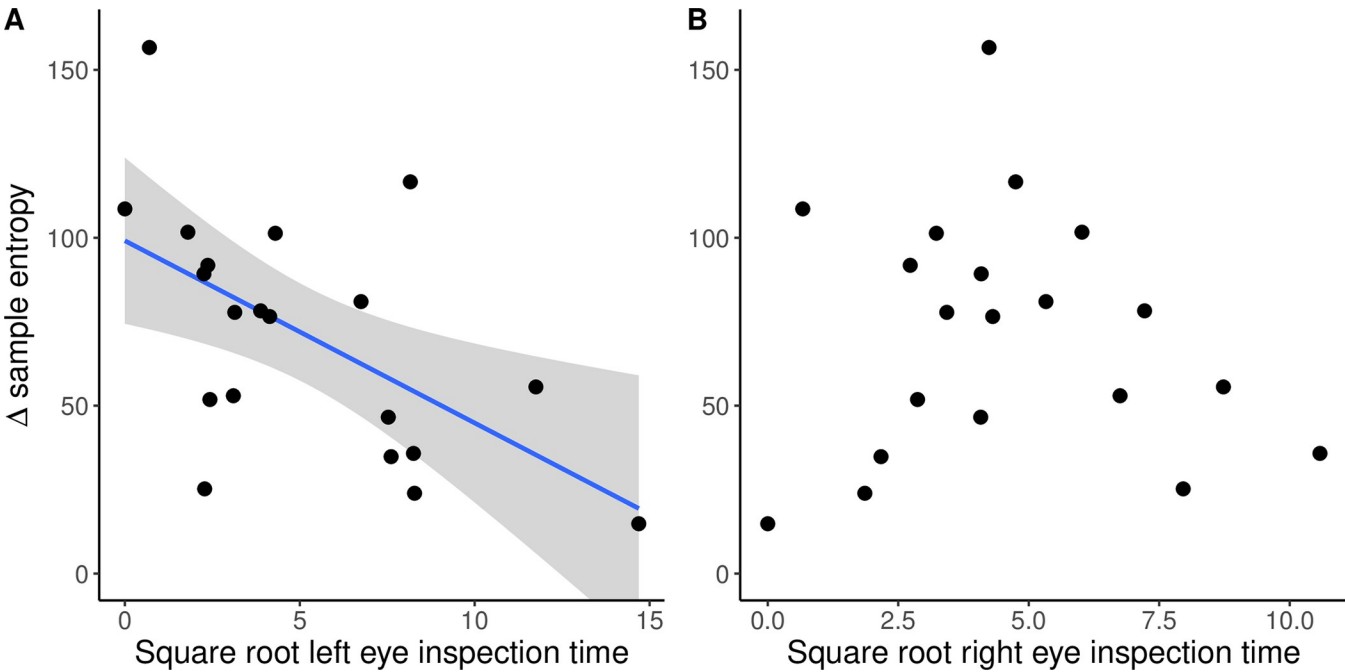

**Fig 2. The linear relationship between the time spent with each eye facing the balloon and the change in sample entropy during the novel object task compared to the baseline measure (n = 20).** A) Left eye inspection time was significantly associated with a reduction in sample entropy (r = -0.57, p = 0.009). The line indicates a least square best fit with 95% confidence interval (shaded region). B) Right eye inspection time was not significantly related to a change in sample entropy (r = -0.08, p = 0.74).

did act as a stress provoking stimulus as anticipated. Most of the horses preferred to inspect the stimulus preferentially with one eye or the other, but the direction of these preferences was not aligned across horses, and 40% of animals showed no clear bias, in agreement with a previous report [1].

The right hemisphere has widely been reported to take precedence in processing frightening or stress-inducing information in the environment [4]. A possible explanation for our results is that horses that perceived the stimulus as more stress provoking, had a larger decrease in SampEn from baseline [25], and thus preferentially processed the information with their right hemisphere [20]. Alternatively, horses that tend to inspect the unfamiliar stimulus with their left eye may be prone to enter emotional states typified by the right hemisphere. Left-eye dominant horses may be more likely to experience fear or stress because the right hemisphere processes the unfamiliar stimulus. In this scenario, laterality phenotype would drive the difference in sample entropy rather than the other way around [26].

Whether underlying laterality drives the emotional response of the horse or whether the emotional response influences eye use bias is related to an open question in the study of emotions: are emotions unconscious peripheral responses of the organism mediated by the ANS and aroused by stimuli in the environment or are they created in specific brain areas and are the consequence of a cognitive process [7, 9, 27]? We detected a possible heart-brain interplay in emotional response to the environment in the horse, however, our current data cannot speak to the causal direction of this association. We chose not to restrain the horses prior to the test to avoid causing additional stress which could obscure or overwhelm any change in SampEn due to the balloon presentation. The balloon was presented at random relative to each horses starting position and therefore it is possible that the eye used primarily to view the balloon and hence the change in SampEn could be determined by the chance positioning of the horse. Our data do not support this perspective as we did not find any differences in left or right eye inspection time, nor ΔSampEn between horses that began the trial with their left, right, or both eyes facing the balloon. Future work in which the starting position of the horse is controlled in a way that does not affect the horse nor interfere with the timing of the ECG data collection would be valuable to help to understand the interplay between lateralised processing and emotional response.

In humans, entropy is a promising marker for mild acute stress [28] and progressively decreases during sympathetic activation, thus offering an alternative measurement of sympathovagal balance [13]. We emphasise the importance of analysing the function of the ANS in assessing the correlation between emotionality and behaviour in equids [18] including in applied areas such as training and the human-horse relationship [29]. In this way the sample entropy could be a useful parameter for assessing an animal's emotionality.

## Conclusion

Lateralised eye use during inspection of a novel object is related to cardiac sample entropy, which may reflect the emotional state of the animal and could explain help to explain the individual variation in lateralisation observed in this species by providing a link between lateralisation and physiology. Clever experimentation will be required to disentangle the causal direction of this relationship, and this would be fertile ground for future work. To our knowledge, our study is the first to find possible heart-brain interplay during an emotional response in horses.

## Supporting information

**S1 Data.**
(XLSX)

## Acknowledgments

We thank the owners of the horses and the staff of the stables involved for their support in the research. We also thank Will Swaney for assistance with Fig 2.

## Author Contributions

**Conceptualization:** Martina Felici, Adam R. Reddon, Veronica Maglieri, Antonio Lanatà, Paolo Baragli.

**Data curation:** Adam R. Reddon, Veronica Maglieri, Paolo Baragli.

**Formal analysis:** Adam R. Reddon, Veronica Maglieri.

**Investigation:** Martina Felici, Antonio Lanatà, Paolo Baragli.

**Methodology:** Antonio Lanatà, Paolo Baragli.

**Project administration:** Paolo Baragli.

**Resources:** Antonio Lanatà, Paolo Baragli.

**Supervision:** Paolo Baragli.

**Visualization:** Adam R. Reddon.

**Writing – original draft:** Martina Felici, Adam R. Reddon, Paolo Baragli.

**Writing – review & editing:** Martina Felici, Adam R. Reddon, Veronica Maglieri, Antonio Lanatà, Paolo Baragli.

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
