## [Decision Letter · Decision Letter 0]

8 Jun 2023

PONE-D-23-13567Heart and brain: Change in cardiac entropy is related to lateralised visual inspection in horsesPLOS ONE

Dear Dr. Reddon,

Thank you for submitting your manuscript to PLOS ONE. After careful consideration, we feel that it has merit but does not fully meet PLOS ONE’s publication criteria as it currently stands. Therefore, we invite you to submit a revised version of the manuscript that addresses the points raised during the review process.

We look forward to receiving your revised manuscript.

Kind regards,

Lesley Joy Rogers, B.Sc. (Hons), D.Phil., D.Sc.

Academic Editor

PLOS ONE

Journal Requirements:

Additional Editor Comments:

Please respond fully to the issues raised by both reviewers.

Reviewers' comments:

Reviewer's Responses to Questions

**Comments to the Author**

1. Is the manuscript technically sound, and do the data support the conclusions?

Reviewer #1: No

Reviewer #2: Partly

2. Has the statistical analysis been performed appropriately and rigorously? 

Reviewer #1: N/A

Reviewer #2: Yes

3. Have the authors made all data underlying the findings in their manuscript fully available?

Reviewer #1: Yes

Reviewer #2: Yes

4. Is the manuscript presented in an intelligible fashion and written in standard English?

Reviewer #1: Yes

Reviewer #2: Yes

5. Review Comments to the Author

Reviewer #1: I read the work with great interest and I found the topic interesting, however, there are several fundamental points of the methodology that leave me perplex on the reliability of the results (please see below).

Results

“The mean LI of horses viewing the balloon was 0.07±0.14, while the mean absolute value of LI was 0.53±0.07, indicating that individual horses tended to show a preferred eye to look at the balloon, but the preference was not aligned with the population level. Out of 20 horses, 7 preferred (≥75% of the time spent in monocular fixation) to use their right eye, 5 preferred their left eye, and 8 showed weak or absent preference between eyes”.

I am not very clear to the calculation of laterality at an individual level: seeing the materials and methods, the calculation of the laterality index is described taking into account the fixation time of the potentially alarm stimulus with a single visual hemi-field. However, this calculation is based on a single test and it is impossible to calculate any type of reliable analysis at the individual level (E.G. One Sample t-Test, verifying that all the trials were statistically significant from the value 0 = equal time spent to fix the target with an eye than to another).

It is not clear to me why the authors wanted to insert the position of the head at the start as a fixed factor in the analysis and did not control the position of the head before insufflating the air in the balloon, even if the horses were not restrained it would have been enough to wait for the horse to be centered compared to the balloon to insufflate the air.

Furthermore, from the visual inspection of the raw data it is evident that the starting position of the horses' head has influenced the laterality index and therefore the results. For this reason, considering the fact that the subjects are not very numerous and that there is the lack of repeated measures, I find that the work is still premature to show scientifically reliable results.

Reviewer #2: This study on changes in cardiac entropy in relation to visual lateralization in horses is a nice follow up of a similar study by the same authors published recently in PloS ONE. Adding a physiological correlate allows to go further into the processes involved in the laterlaization observed when horses are confronted to a novel object. Since the protocol is identical, these complementary informations are particularly welcome. I just wondered whether the proposed study could have been part of the previous one (some horses equipped for ECG) in which case this should be mentioned. The physiological measure is original and provides an interesting new tool for "ranking" the animals according to cardiac changes from baseline to test situation. Looking at the results I wondered whether the laterality index was the best measure for testing the relationship with the cardiac entropy. On one hand, the LI indicates that almost half of the horses show no laterality bias, on the other hand it is correlated with cardiac entropy. I would suggest to test and possibly add the correlation between the time spent gazing at the object (or having it in the hemifield, which is not quite the same and needs to be clarified) with the left (and right respectively) eye and the cardiac entropy. This may be still clearer than the LI. Larose et al. (2006) found a correlation between the time spent with the LE towards a novel object and their index of emotionality based on behavioural measures.Since horses can also be looking elsewhere, the time spent with one given eye towards the object may well bring some additional information as compared to LI. I would not necessarily consider the hemifield at the start of the balloon inflation as a result, but rather as a limitation. It is probably difficult to determine at which stage precisely of the inflation, which although rapid, requires at least a few seconds, the stimulus is perceived as frightening. There should have been more control over when to start inflating according to the horse's position in the box, although I agree it may be difficult to achieve. It would have been nice also to have other behavioural measures such as startles, flight, snores (see Scopa et al. 2018, Contrerar-Aguilas et al. 2019) to have a more complete view of the emotional state of the horses and bring elements on your discussion on heart/brain and what comes first. D'Ingeo et al. 2019 explain their results on laterality by a first immédiate responses (e.g. ear reflex) and then analysis of the situation (EEG data, other hemisphere). More complete behavioural analyses would probably enrich further this discussion.

Just a few additional details:

Why are there only 16 horses in Figure 2?

Line 112: what do you mean by "more pronounced stereotypies"? This does not make much sense: either you really observed the behaviour of the horses in their box outside the tests with appropriate sampling methods or you just can not know, nor is it clear whether some tyes of stereotypies are functionally more important than others. I just would not mention any indicator of welfare if not measured appropriately with other indicators as well.

The data are made available in the suppl. material part but I was a bit frustrated that the real data on the time spent with the object in the different hemifields was not there

I wonder whether a parallel with Sankey et al 2010 's results showing both a left eye use and elevated heart rate in horses trained with negative reinforcement (as compared to positive rft) could be interesting to discuss

6. PLOS authors have the option to publish the peer review history of their article (what does this mean?). If published, this will include your full peer review and any attached files.

Reviewer #1: No

Reviewer #2: No

---

## [Author Response · Author response to Decision Letter 0]

3 Jul 2023

Reviewer #1:

I am not very clear to the calculation of laterality at an individual level: seeing the materials and methods, the calculation of the laterality index is described taking into account the fixation time of the potentially alarm stimulus with a single visual hemi-field. However, this calculation is based on a single test and it is impossible to calculate any type of reliable analysis at the individual level (E.G. One Sample t-Test, verifying that all the trials were statistically significant from the value 0 = equal time spent to fix the target with an eye than to another).

The reviewer is correct, we cannot say that our horses are not lateralized in general based on a single test. We have deemphasized the laterality index in general in response to reviewer 2’s suggestion about how to analyse the data based on the use of each eye and include it here only to facilitate comparison with previous work. We have also emphasised that the eye use preferences may be either a cause or consequence of the emotional response to the stimulus. 

It is not clear to me why the authors wanted to insert the position of the head at the start as a fixed factor in the analysis and did not control the position of the head before insufflating the air in the balloon, even if the horses were not restrained it would have been enough to wait for the horse to be centred compared to the balloon to insufflate the air. Furthermore, from the visual inspection of the raw data it is evident that the starting position of the horses' head has influenced the laterality index and therefore the results.

We chose not to restrain the horses as this could itself lead to stress and swamp out the effects we were looking for (see Yarnel et al., 2013; Vitale et al., 2013; Mograbi et al., 2020). Furthermore, it is not a simple decision to merely wait until the horse is facing the right way to start the trial, as this would affect the timings of the ECG data collection, introducing a different uncontrolled variable. We do include greater discussion of the potential limitations imposed by our lack of control over the starting position (Lines 155-159; 206-215) and include some new analyses comparing ΔSampEn and time spent using each eye between horses that started with their left, right, or both eyes facing the balloon (Line 172-175). We did not find any group level differences in any of these parameters, which although the power to detect these differences is limited, does suggest that starting position does not have a major effect on the responses we collected. 

Yarnell, K., Hall, C., & Billett, E. (2013). An assessment of the aversive nature of an animal management procedure (clipping) using behavioral and physiological measures. Physiology & behavior, 118, 32-39; 

Vitale, V., Balocchi, R., Varanini, M., Sgorbini, M., Macerata, A., Sighieri, C., & Baragli, P. (2013). The effects of restriction of movement on the reliability of heart rate variability measurements in the horse (Equus caballus). Journal of Veterinary Behavior, 8(5), 400-403;

Mograbi, K. D. M., Suchecki, D., da Silva, S. G., Covolan, L., & Hamani, C. (2020). Chronic unpredictable restraint stress increases hippocampal pro-inflammatory cytokines and decreases motivated behavior in rats. Stress, 23(4), 427-436.).

Reviewer #2: 

This study on changes in cardiac entropy in relation to visual lateralization in horses is a nice follow up of a similar study by the same authors published recently in PloS ONE. Adding a physiological correlate allows to go further into the processes involved in the laterlaization observed when horses are confronted to a novel object. Since the protocol is identical, these complementary informations are particularly welcome. I just wondered whether the proposed study could have been part of the previous one (some horses equipped for ECG) in which case this should be mentioned. The physiological measure is original and provides an interesting new tool for "ranking" the animals according to cardiac changes from baseline to test situation. 

We thank the reviewer for their positive assessment of our work. This study used and entirely new sample of horses, naïve to the task from the previous report. We have made this clear in the manuscript (Line 138) when talking about comparing the studies. 

Looking at the results I wondered whether the laterality index was the best measure for testing the relationship with the cardiac entropy. On one hand, the LI indicates that almost half of the horses show no laterality bias, on the other hand it is correlated with cardiac entropy. I would suggest to test and possibly add the correlation between the time spent gazing at the object (or having it in the hemifield, which is not quite the same and needs to be clarified) with the left (and right respectively) eye and the cardiac entropy. This may be still clearer than the LI. Larose et al. (2006) found a correlation between the time spent with the LE towards a novel object and their index of emotionality based on behavioural measures. Since horses can also be looking elsewhere, the time spent with one given eye towards the object may well bring some additional information as compared to LI. 

We thank the reviewer for this excellent suggestion. We have redone the analysis to focus on the correlation between the time spent with each eye facing the novel object and the change in sample entropy. This new analysis shows a clear relationship between the time with the left eye facing the object and the reduction in entropy observed, while the time spent using the right eye was unrelated. We believe this change in analysis shows our effect more clearly and represents a significant improvement for the paper. See the results and new figure 2 for more information. We do still include the LI calculation and description of the sample breakdown to facilitate comparison with our previous paper and other related literature, but we no longer focus on the laterality index in the current manuscript. 

I would not necessarily consider the hemifield at the start of the balloon inflation as a result, but rather as a limitation. It is probably difficult to determine at which stage precisely of the inflation, which although rapid, requires at least a few seconds, the stimulus is perceived as frightening. There should have been more control over when to start inflating according to the horse's position in the box, although I agree it may be difficult to achieve. It would have been nice also to have other behavioural measures such as startles, flight, snores (see Scopa et al. 2018, Contrerar-Aguilas et al. 2019) to have a more complete view of the emotional state of the horses and bring elements on your discussion on heart/brain and what comes first. D'Ingeo et al. 2019 explain their results on laterality by a first immédiate responses (e.g. ear reflex) and then analysis of the situation (EEG data, other hemisphere). More complete behavioural analyses would probably enrich further this discussion.

 We have now emphasised that the lack of control over starting position is a limitation for our work (lines 156-159; 206-215). We have added some new analysis showing that starting position doesn’t have a strong influence on eye use or change in entropy, but nevertheless, future work should consider how best to treat the starting position of the animal during this type of task. We thank the reviewer for their excellent suggestions for future directions as well. 

Just a few additional details:

Why are there only 16 horses in Figure 2?

Initially we excluded the frontal facing horses from this figure because they were not included in the ANCOVA analysis. However, as the reviewer suggests we have changed the analysis to focus on the time spent looking with each eye separately, and we have emphasised that our lack of control over starting position is a limitation rather than a factor of interest (on the suggestion of both reviewers), and therefore the updated figure 2 includes all 20 horses. 

Line 112: what do you mean by "more pronounced stereotypies"? This does not make much sense: either you really observed the behaviour of the horses in their box outside the tests with appropriate sampling methods or you just can not know, nor is it clear whether some tyes of stereotypies are functionally more important than others. I just would not mention any indicator of welfare if not measured appropriately with other indicators as well.

Stereotyped behaviours may affect cognition and reaction to stimuli so we decided to have stereotyped behaviours a criteria for exclusion. However, no horses were excluded on this basis, and rather than get into the details of welfare assessments which were not the focus of this paper, we have done as the reviewer suggests a removed this mention of stereotypies. 

The data are made available in the suppl. material part but I was a bit frustrated that the real data on the time spent with the object in the different hemifields was not there

I wonder whether a parallel with Sankey et al 2010 's results showing both a left eye use and elevated heart rate in horses trained with negative reinforcement (as compared to positive rft) could be interesting to discuss

We have redone the analysis to focus on time spent looking with each eye rather than the laterality index as suggested by the reviewer. We have included this raw data on time spent looking with each eye in our data file in the supplementary material. We have added the Sankey et al (2010) paper to our discussion (Line 221) and thank the reviewer for alerting us to this relevant reference.

---

## [Decision Letter · Decision Letter 1]

25 Jul 2023

Heart and brain: Change in cardiac entropy is related to lateralised visual inspection in horses

PONE-D-23-13567R1

Dear Dr. Reddon,

We’re pleased to inform you that your manuscript has been judged scientifically suitable for publication and will be formally accepted for publication once it meets all outstanding technical requirements.

Kind regards,

Lesley Joy Rogers, B.Sc. (Hons), D.Phil., D.Sc.

Academic Editor

PLOS ONE

Additional Editor Comments (optional):

Thank you for revising your paper. You have addressed all of the reviewers' concerns and the paper will now make an important contribution to the field.

Reviewers' comments:

Reviewer's Responses to Questions

**Comments to the Author**

1. If the authors have adequately addressed your comments raised in a previous round of review and you feel that this manuscript is now acceptable for publication, you may indicate that here to bypass the “Comments to the Author” section, enter your conflict of interest statement in the “Confidential to Editor” section, and submit your "Accept" recommendation.

Reviewer #1: All comments have been addressed

Reviewer #2: All comments have been addressed

2. Is the manuscript technically sound, and do the data support the conclusions?

Reviewer #1: Yes

Reviewer #2: Yes

3. Has the statistical analysis been performed appropriately and rigorously? 

Reviewer #1: Yes

Reviewer #2: Yes

4. Have the authors made all data underlying the findings in their manuscript fully available?

Reviewer #1: Yes

Reviewer #2: Yes

5. Is the manuscript presented in an intelligible fashion and written in standard English?

Reviewer #1: Yes

Reviewer #2: Yes

6. Review Comments to the Author

Reviewer #1: (No Response)

Reviewer #2: The authors have responded adequately to all my comments and I am glad that this helped the results become still clearer. I believe the manuscript is ready for publication

7. PLOS authors have the option to publish the peer review history of their article (what does this mean?). If published, this will include your full peer review and any attached files.

Reviewer #1: No

Reviewer #2: No

---

## [Editor Report · Acceptance letter]

28 Jul 2023

PONE-D-23-13567R1 

Heart and brain: Change in cardiac entropy is related to lateralised visual inspection in horses 

Dear Dr. Reddon:

I'm pleased to inform you that your manuscript has been deemed suitable for publication in PLOS ONE. Congratulations! Your manuscript is now with our production department. 

Kind regards, 

on behalf of

Prof. Lesley Joy Rogers 

Academic Editor

PLOS ONE